# Age-Dependent Impact of Concomitant Radio-Chemotherapy and MGMT Promotor Methylation on PFS and OS in Patients with IDH Wild-Type Glioblastoma: The Real-Life Data

**DOI:** 10.3390/cancers14246180

**Published:** 2022-12-14

**Authors:** Aleksandrs Krigers, Julia Klingenschmid, Tolga Cosar, Patrizia Moser, Claudius Thomé, Christian F. Freyschlag

**Affiliations:** 1Department of Neurosurgery, Medical University of Innsbruck, Anichstrasse 35, 6020 Innsbruck, Austria; 2Department of Neuropathology, Tirol Kliniken, Anichstrasse 35, 6020 Innsbruck, Austria

**Keywords:** glioblastoma, age, elderly, MGMT promotor methylation, concomitant radio-chemotherapy, PFS, OS

## Abstract

**Simple Summary:**

Nowadays: biological but not chronological age and performance have more influence on decision making by patients with malignant brain tissue tumors. We showed how more aggressive therapy results in the increased life expectancy of older patients with this disease in real life. More than half of all patients with these tumors were older than 65 years of age. Even if survival was shorter than in the younger patients, aggressive radio-chemotherapy after the surgery also provided additional time for seniors. In real life, genetically favorable features of the tumor had a positive influence on survival only if more aggressive therapy was applied.

**Abstract:**

Biological but not chronological age plus performance have more impact on decision making in glioblastoma patients. We investigated how progression-free survival (PFS) and overall survival (OS) in older patients with IDH wild-type glioblastoma were influenced by concomitant radio-chemotherapy and MGMT promotor methylation status in real-life settings. In total, 142 out of 273 (52%) evaluated patients were older than 65 years, and 77 (55%) of them received concomitant radio-chemotherapy. In senior patients, the initiation of concomitant radio-chemotherapy was associated with significantly better PFS: 15.3 months (CI95: 11.7–18.9) vs. 7.0 months (CI95: 4.3–9.6; *p* = 0.002). The favorable influence on PFS was not related to MGMT promotor methylation status as it was in the younger cohort. In seniors, concomitant radio-chemotherapy was related to significantly better OS: 20.0 months (CI95: 14.3–26.7) vs. 4.9 months (CI95: 3.5–6.2), *p* < 0.001. MGMT promotor methylation was related to a more favorable OS only, if concomitant radio-chemotherapy was initiated. In conclusion, more than half of the glioblastoma cohort was older than 65 years of age. Even if PFS and OS were shorter than in the younger cohort, concomitant radio-chemotherapy provided a survival advantage. In real life, MGMT promotor methylation had a positive impact on OS only if the adjuvant therapy was applied.

## 1. Introduction

Glioblastoma is the most frequent primary malignant tumor of the central nervous system (CNS) [1], and its incidence is rising [2]. This could be explained by better availability of cranial imaging [3] and increased life expectancy [4]. Nevertheless, outcome remains poor due to limited therapeutic options [5,6,7].

According to the actual Central Brain Tumor Registry of the United States (CBTRUS) report, there is a trend for increased patients’ age in firstly diagnosed glioblastoma, reaching a peak between 75 to 84 years [1]. Age was also described as an independent negative prognostic factor in patients with glioblastoma [8,9]. Thus, a higher number of glioblastoma cases in an elderly population including the unfavorable prognosis is to be expected in clinical practice.

Glioblastomas are not homogenous: their behavior, as well as the efficacy of targeted treatment and consequently outcome is associated with glioblastoma characteristics. Nowadays, only prognostically more unfavorable isocitrate dehydrogenase (IDH) wild-type World Health Organization (WHO) grade 4 gliomas are taxonomically considered as glioblastoma [10]. At the same time, another epigenetic alteration—O6-Methylguanine-DNA-Methyltransferase (MGMT) promotor methylation is known to be associated with a better treatment response and outcome. According to Stupp et al., overall survival (OS) after standardized adjuvant first-choice concomitant radio-chemotherapy in cases where MGMT promotor methylation was 23.4 months, compared to only 12.6 months otherwise [11,12]. 

Nevertheless, combined radio-chemotherapy has been described as incriminating for older patients to tolerate. Thus, in cases of MGMT promotor methylation, temozolomide (TMZ) monotherapy, or alternatively, hypo-fractioned radiotherapy have been suggested [13,14]. However, these trials were performed with conservative age cut-offs, low MGMT assessment quote and the previous WHO classification. The outcome in cases of adjuvant monotherapy remained worse compared to a standard treatment regimen. Furthermore, patients older than 65 years are generally underrepresented in neuro-oncological studies [15].

Nowadays, biological but not chronological age in combination with frailty have more impact on the decision making in cases of malignant glioma [16,17]. Thus, our center generally offers aggressive tumor resection and concomitant radio-chemotherapy to elderly patients with low frailty. We aimed to investigate how progression-free survival (PFS) and OS in older patients with IDH wild-type glioblastoma are impacted by concomitant radio-chemotherapy and MGMT promotor methylation status in real-life settings.

## 2. Materials and Methods

All adult patients with IDH wild-type glioblastoma who were operated on for the first time in our center from 2012 to 2021 were selected. Tumors classified as IDH-mutated glioblastoma by neuropathological examination were excluded. Hence, this provided a representable taxonomic cohort in accordance with the current 5th WHO classification of tumors of the central nervous system from 2021 [10]. Cases with recurrent pathologies were also excluded.

Epidemiological, neuropathological and clinical information was retrospectively collected from an institutional database. The preoperative Eastern Co-operative of Oncology Group (ECOG) or Karnofsky Performance Score (KPI) score was noted as clinical standard [18]. Maximal safe resection was the first-choice treatment, and if this was not possible, a navigated biopsy was performed. Standardized neuropathological assessment of FFP-embedded tissue was carried out in each individual case. Glioma grading was carried out respecting the WHO classification of tumors of the central nervous system. According to the 4th edition from 2007 and the revised 4th edition from 2016, the same glioblastoma definition based on phenotypical features such as necrosis and microvascular hyperproliferation was applied to all cases [19,20]. IDH1 was evaluated by immunohistochemistry (IHC) to reveal R132H mutation. In cases of absent specific expression, DNA sequencing was followed for patients under 40 years of age in order to confirm wild-type IDH. MGMT promotor methylation status was defined as more than a mean of 8% methylation in 4 analyzed CpG gen areas in pyrosequencing.

Adjuvant treatment of each patient was individually discussed in a multidisciplinary tumor board, in which biological age (as opposed to chronological age) and frailty were taken into account for performance evaluation. The concomitant radio-chemotherapy with adjuvant 6 cycles of TMZ monotherapy according to Stupp et al. was the first suggestion [11,12]. For frail patients with a KPI from 50 to 70, lower-dose radiation therapy or TMZ monotherapy in case of MGMT promotor methylation was considered. If KPI was lower than 50 without a realistic perspective for its recovery, the best supportive care was proposed [21,22,23]. Routine follow-up was arranged every 3 months. The outcome end-points of the study were defined as PFS and OS after the surgery. Oncological progression was defined according to the Response Assessment in Neuro-Oncology (RANO) criteria [6]. If PFS and/or OS outcome were not reached or accessible, the last available contact timepoint was noticed.

Data processing, statistical evaluation and figure creation were conducted using IBM SPSS Statistics (IBM SPSS Statistics for Mac OS, Version 27.0. Armonk, NY, USA: IBM Corp.). Comparison of binominal variable pairs was performed with Chi-squared test. The mean estimated PFS and OS times were acquired with Kaplan–Meier processing and demonstrated with survival graphs. The influence of univariate binominal variables on PFS and OS was compared using the LogRank test. Cox regression analysis was applied to reveal hazard ratios for oncological progression or death during follow-up. For dichotomic sub-group comparison considering age, patients were divided into a younger (18 to 65 years) and older cohort (66 years and older) according to WHO standard, previous studies [13,15] and internal mean. The α value was 0.05, and 95% confidence intervals were constructed. 

This study was conducted in accordance with the Declaration of Helsinki and its future amendments. Ethical approval was obtained (EK Nr: 1333/2021, Medical University of Innsbruck).

## 3. Results

In total, 273 patients with firstly operated IDH wild-type glioblastoma were evaluated. Mean age of the 155 (57%) males and 118 (43%) females was 64 years (CI95: 62–66). The highest preoperative performance score (ECOG = 0 s. KPI = 90–100) was present in 175 (64%) cases. Overall, 214 (79%) patients received tumor resections and another 58 (21%) patients only had biopsies. The MGMT promotor was methylated in 116 (48%) cases and remained operational in 125 (52%) tumors, its status was not assessed in 32 cases. 

More than half of the patients (142, 52%) were older than 65 years. They showed the highest preoperative performance score (ECOG = 0) less frequently than the younger group: 69 (59%) vs. 106 (81%) patients correspondingly (*p* < 0.001). Biopsy instead of resection was performed more often in the older cohort compared to the younger one: 41 (29%) vs. 17 (13%) cases respectively (*p* = 0.001). MGMT promotor methylation was confirmed in 42% of cases (47/113) in the younger cohort and 54% (69/128) in the older cohort. The tumor-board selected adjuvant treatment is provided in Table 1. TMZ was not applied in 13 cases in the younger cohort including 9 due to low performance, 2 due to the patient’s personal choice, 1 due to known thrombotic thrombocytopenic purpura and 1 due to pregnancy. TMZ was not started in 49 patients of the older cohort, including 39 with low performance, 6 due to the patient’s personal choice and 1 with known idiopathic thrombocytopenia.

### 3.1. PFS in the Entire Cohort

Mean follow-up in our study was 12.9 months (CI95: 11.2–14.6). During follow-up, oncological progression was described in 152 (57%) cases. In the entire cohort, mean estimated PFS was 18.1 months (CI95: 15.3–21.0).

In cases of MGMT promotor methylation, mean PFS was 22.5 months (CI95: 17.1–27.8) vs. 13.0 months, if it remained active (CI95: 10.6–15.4, *p* = 0.001). If biopsy was performed instead of resection, mean PFS was shorter: 8.5 months (CI95: 4.8–12.2) vs. 19.0 months (CI95: 15.9–22.0), respectively (*p* = 0.032). At the same time, the highest preoperative performance (ECOG = 0) compared to lower scores was not associated with PFS changes. PFS in accordance with MGMT promotor methylation status, surgical modality and preoperative performance is shown in Appendix A Figure A1A–C as Kaplan–Meier graphs.

Independent hazard ratio (HR) for oncological progression during follow-up in cases of unmethylated MGMT promotor was 1.77 [(CI95: 1.24–2.53), variable *p* = 0.002, regression *p* = 0.001]. If only biopsy was performed, HR for oncological progression was 2.16 [(CI95: 1.28–3.66), variable *p* = 0.003, regression *p* = 0.004]. Preoperative ECOG performance score was not associated with PFS changes according to Cox regression. 

If adjuvant concomitant radio-chemotherapy was initiated, mean PFS was 19.4 months (CI95: 16.8–22.5) vs. 8.4 months (CI95: 5.3–11.6) otherwise (*p* < 0.001), whereas the statistically significant differences were noticed only in cases of methylated MGMT promotor: mean PFS was 25.0 months (CI95: 19.2–30.9) vs. 8.3 months (CI95: 4.2–12.5), if the MGMT promotor stayed active (*p* < 0.001, s. Figure 1A,B).

### 3.2. OS in the Entire Cohort

A total of 180 (66%) patients reached OS during our follow-up. Mean estimated OS was 20.4 months (CI95: 16.9–23.9) according to Kaplan–Meier processing.

If MGMT promotor was methylated, mean OS was 25.7 months (CI95: 19.7–31.7) or 14.0 months (CI95: 11.6–16.5) if it remained unmethylated (*p* = 0.009). In cases of biopsy only, mean OS was 5.4 months (CI95: 3.9–6.9) compared to 24.0 months (CI95: 19.9–28.1) in cases of resection (*p* < 0.001). If patient preoperatively was in an excellent condition (ECOG = 0), mean OS was 23.1 months (CI95: 18.4–27.8), otherwise mean OS was significantly lower or 15.5 months (CI95: 11.0–19.9, *p* = 0.005). OS development in relation to MGMT promotor methylation status, surgical modality and preoperative performance is demonstrated in Appendix A Figure A1D–F as Kaplan–Meier graphs.

Hazard ratio for the deceased during study follow-up in cases of unmethylated MGMT promotor was 1.53 [(CI95: 1.11–2.11), variable *p* = 0.009, regression *p* = 0.010]. If only biopsy was performed, HR for the deceased was 4.83 [(CI95: 3.33–7.604, variable *p* < 0.001, regression *p* < 0.001] or significantly higher compared to resection. Reduced performance score (ECOG ≥ 1) was associated with less favorable hazards considering OS [HR = 1.52 (CI95: 1.14–2.09), variable *p* = 0.005, regression *p* = 0.005].

If adjuvant concomitant radio-chemotherapy was initiated, mean OS was 25.8 months (CI95: 21.4–30.3) vs. 5.1 months (CI95: 3.8–6.4) otherwise (*p* < 0.001). The statistically significant differences were noticed independently from MGMT promotor status. If it was methylated, mean OS was 36.6 months (CI95: 28.7–44.6) after concomitant radio-chemotherapy vs. 4.9 months (CI95: 3.1–6.7, *p* < 0.001). If MGMT remained operational, mean OS was 16.1 months (CI95: 13.3–18.9) in cases of adjuvant radio-chemotherapy vs. 5.8 months (CI95: 3.5–8.1, *p* < 0.001; Figure 1C,D). 

### 3.3. Age-Dependent Influence on PFS

Mean follow-up for the younger group (18–65 years) was 17.8 months (CI95: 14.9–20.7) and 8.4 months (CI95: 6.7–10.0) for the older group (>65 years).

Mean estimated PFS in the younger cohort was 21.1 months (CI95: 17.1–25.0) vs. 13.1 months (CI95: 10.1–16.1) in the older one (*p* = 0.003). In senior patients, initiated concomitant radio-chemotherapy was associated with significantly longer PFS: 15.3 months (CI95: 11.7–18.9) vs. 7.0 months (CI95: 4.3–9.6), *p* = 0.002. At the same time, the influence of concomitant radio-chemotherapy on PFS was not related to MGMT promotor methylation status as it was for the younger cohort (Table 2 and Figure 2A–C). 

The data considering PFS for younger (18 to 65 years) and older patients (more than 65 years) in relation to MGMT promotor methylation status, surgical modality and preoperative ECOG are shown in Table 2.

Corresponding results were revealed in Cox regression modeling. Patients older than 65 years of age showed significantly better PFS hazards during follow-up, if combined radio-chemotherapy was initiated [HR = 2.42 (CI95: 1.37–4.19), variable *p* = 0.002, regression *p* = 0.002]. On the other hand, the hazard ratio for oncological progression in cases of MGMT promotor methylation for younger patients was 2.54 [(CI95: 1.55–4.18), variable *p* < 0.001, regression *p* < 0.001], and there were no statistically significant differences for older patients. Preoperative performance score or surgical modality did not provide additional PFS hazards in both age groups in accordance with Cox regression. 

### 3.4. Age-Dependent Influence on OS

Mean estimated OS in the younger cohort was longer than in the older cohort: 27.6 months (CI95: 22.0–33.1) vs. 13.6 months (CI95: 10.0–17.2) respectively (*p* < 0.001). 

In patients older than 65 years, concomitant radio-chemotherapy was related to significantly better survival: 20.0 months (CI95: 14.3–26.7) vs. 4.9 months (CI95: 3.5–6.2, *p* < 0.001). The same association was found for younger patients: mean OS was 29.8 months (CI95: 23.9–35.8) vs. 6.1 months (CI95: 2.1–10.2) respectively (*p* < 0.001; Figure 2D–F). MGMT promotor methylation was relevant for outcome only if concomitant radio-chemotherapy was initiated (Table 3). OS in relation to MGMT promotor methylation status, surgical modality and preoperative ECOG for both cohorts is provided in Table 3.

Same results were revealed in Cox regression modeling. The older cohort showed significantly better hazards to survive during follow-up if radio-chemotherapy was initiated [HR = 2.42 (CI95: 1.37–4.19), variable *p* < 0.001, regression *p* < 0.001]. The hazard ratio for the deceased if MGMT promotor remained operational was 4.18 for older patients [(CI95: 2.75–6.37), variable *p* < 0.001, regression *p* < 0.001], and 6.67 for younger patients [(HR = CI95: 3.41–13.0), variable *p* < 0.001, regression *p* < 0.001].

Additionally, biopsy instead of resection was associated with higher hazards for death in older patients [(HR = 3.57, CI95: 2.27–5.59), variable *p* < 0.001, regression *p* < 0.001], and younger patients [HR = 5.78 (CI95: 2.84–11.24), variable *p* < 0.001, regression *p* < 0.001]. Preoperative performance score did not provide additional OS hazards in either age-groups.

## 4. Discussion

Real-life PFS and OS was evaluated in a large cohort of 273 firstly diagnosed IDH wild-type glioblastomas. More than half were older than 65 years of age, whereas 55% of those patients received concomitant radio-chemotherapy, resulting in statistically significant better PFS and OS. MGMT promotor methylation was not associated per se with better survival in the older cohort. Only if concomitant radio-chemotherapy was initiated, it resulted in better survival in cases of methylated MGMT promotor.

Generally, the population is getting older and the disability-free life span is increasing [4]. Nevertheless, the definition of the term ‘elderly’ remains variable: the WHO defines it as above 65 years old and miscellaneous cut-offs are used in the clinical and scientific field [24]. The glioblastoma prevalence in the older population is also increasing. In 2030, 70% of all malignancies will be found in patients older than 65 years [25]. As our data demonstrated, for glioblastoma IDH wild-type, this has already been the case. The peak incidence for malignant glioma is increasing and has already reached more than 75 years [1]. At the same time, older patients have been commonly excluded from the oncological trials [15]. 

The treatment of choice in newly diagnosed glioblastoma is neurologically safe maximum tumor resection, followed by concomitant radio-chemotherapy [11]. Instead, TMZ monotherapy in cases of MGMT promotor methylation, or otherwise, hypo-fractioned radiotherapy has been advised for older patients [13,14]. Following those recommendations, more than half of the patients in our cohort should have received any kind of reduced treatment. Despite this, we looked for biological age and co-morbidities as a part of performance and frailty assessment in our decision making. Thus, more than half of the patients older than 65 years received intensive treatment.

According to our data, resection instead of biopsy is associated with longer PFS and OS in the entire cohort. This data are concordant with the literature [16,26,27]. In our sub-group analysis, patients older than 65 years of age showed additional 12.4 months of survival, if resection instead of biopsy only was performed. The same was determined for younger patients with 22.6 months’ increased OS. Thus, neurologically safe maximal tumor resection should be favored in all patients due to significantly longer OS and a trend for longer PFS.

MGMT promotor methylation is a prognostically favorable factor [28,29]. The explanation lies in the ability of the MGMT enzymatic product to overhaul the alkylating effect of TMZ. In cases of epigenetic inhibition of the MGMT gene, translation and further synthesis of O-6-methylguanine-DNA-methyltransferase is limited. Due to the reduced quantity of this enzyme, the DNA repair, e.g., reversing of TMZ-induced modification O-6-methylguanine back to guanine is not performed, resulting in cellular apoptosis [29]. Thus, MGMT promotor methylation can realize its favorable impact only if TMZ or another alkylating agent is applied. This explains, why the methylated MGMT promotor was not associated with a PFS or OS benefit by its sole presence in our older patient cohort, where only two-thirds patients received TMZ. 

On the other hand, if concomitant radio-chemotherapy with TMZ was initiated, it provided survival benefits for older patients in our cohort: 8.3 months longer PFS and 15.1 months longer OS. Furthermore, if concomitant radio-chemotherapy was initiated for senior patients and the MGMT promotor was methylated, OS was prolonged by 14.9 months compared to cases with unmethylated MGMT promotor. Radio-chemotherapy influence on PFS was not related to MGMT promotor methylation status like it was for the younger cohort, but an analogous trend was noticed. At the same time, the combined group of TMZ monotherapy or radio-chemotherapy failed to show any PFS or OS benefits in relation to MGMT promotor methylation. Thus, combined radio-chemotherapy should be the first choice for suitable older patients.

In comparison to historic cohorts, survival data in our study seemed superior. It could be potentially explained by advances in surgical techniques, as well as by a more liberal selection of patients for concomitant radio-chemotherapy disregarding chronological age as a strict contraindication. The median event-free survival of patients older than 65 years of age in the study of Wick et al. was 4.7 months in the RTX group, 3.3 months in the TMZ group and in cases of patients with MGMT promotor methylation who received TMZ, 8.4 months. According to our data, mean PFS in patients for a comparable cohort (>65 years) was 13.1 months. In cases of applied concomitant radio-chemotherapy, this was 15.3 months for all older patients, including 18.6 months if the MGMT promotor was methylated and 11.5 months if the MGMT promotor remained active.

Nevertheless, mean PFS in the younger group (18–65 years) was found to be 8 months longer and OS 24 months longer than in older cohort (>65 years), even if the more intense concomitant radio-chemotherapy was applied. This could be explained with frailty and significant co-morbidities in the older population. Mean OS for the entire cohort was 20.4 or comparable with 20.9 months according to recent data [30], whereas in this study, all eligible patients were included and no restricted exclusion criteria were applied. 

Our study has limitations including the retrospective design that could lead to incomplete data. On the other hand, our standardized clinical documentation allowed to compensate for this concern. We did not evaluate pre-existing specific co-morbidities separately, in order to maintain data quality by retrospective design. Nevertheless, only patients who were suitable for intracranial surgery under general anesthesia were selected. Thus, the terminally ill were not included and did not interfere our results. Further prospective validation is necessary.

## 5. Conclusions

In real-life settings, more than half of our glioblastoma patients were older than 65 years of age. Even if PFS and OS in this cohort remain worse than in younger patients, concomitant radio-chemotherapy delivered survival advantages. The methylated MGMT promotor’s favorable impact on OS could be used if concomitant treatment was applied. Biological and not chronological age including frailty should be valued in decision making in elder patients with GBM.

## Figures and Tables

**Figure 1 cancers-14-06180-f001:**
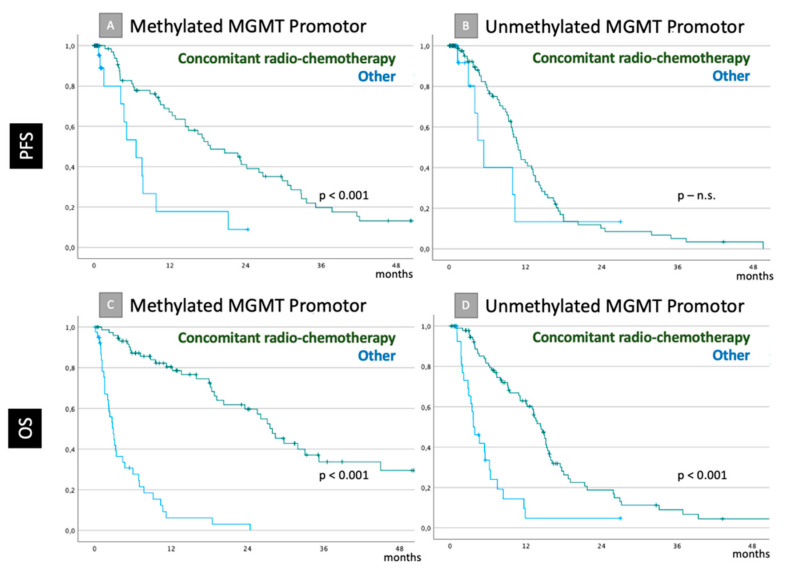
PFS (**A**,**B**) and OS (**C**,**D**) in the entire cohort considering MGMT promotor status if concomitant radio-chemotherapy was initiated or not is shown as Kaplan–Meier curves. Significance level is provided in conformity to LogRank tests. Statistically significant differences were revealed in all cases (**A**,**C**,**D**), except for PFS changes in case of an unmethylated MGMT promotor (**B**).

**Figure 2 cancers-14-06180-f002:**
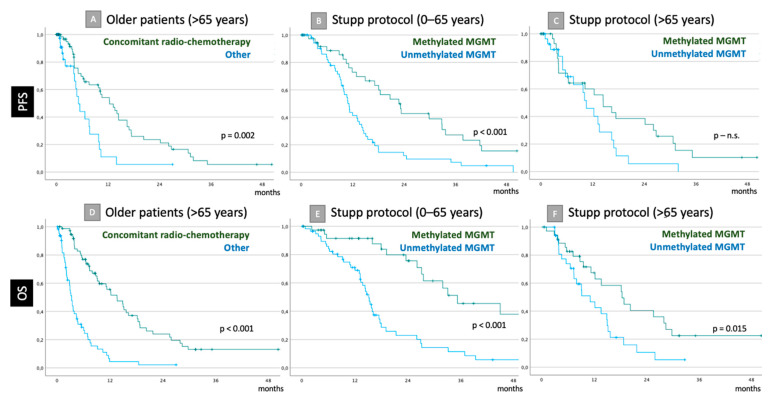
Kaplan–Meier graphs for PFS (**A**–**C**) and OS (**D**–**F**). For the older cohort (>65 years), if concomitant radio-chemotherapy was initiated or not (**A**,**D**). If concomitant radio-chemotherapy was applied, survival differences were presented in association with MGMT promotor status (MGMT) for younger (**B**,**E**) and older patients (**C**,**F**). Significance level is provided in conformity to LogRank tests. Statistically significant differences were revealed for all pairs except for concomitant radio-chemotherapy efficacy in relation to MGMT promotor status in older cohort (**C**).

**Table 1 cancers-14-06180-t001:** Initiated adjuvant therapy for younger (18 to 65 years) and older patients (more than 65 years).

Therapy	Younger Patients (18–65 Years)	Older Patients (>65 Years)
Concomitant TMZ + RTX	88% (116/131)	55% (77/142)
TMZ alone	2% (2/131)	13% (19/142)
Radiotherapy alone	3% (4/131)	7% (10/142)
Best supportive care	5% (7/131)	21% (30/142)
No adjuvant therapy due to the patients’ decision	2% (2/131)	4% (6/142)

TMZ—temozolomide, RTX—radiotherapy.

**Table 2 cancers-14-06180-t002:** Mean estimated PFS in comparison between older (more than 65 years) and younger patients (18 to 65 years) in relation to MGMT promotor methylation status, adjuvant therapy, surgical modality and preoperative ECOG.

Variable		18–65 Years, OS in Months	*p*	>65 Years, OS in Months	*p*
MGMT Promotor	Methylated	28.8 (CI95: 21.1–36.6)	<0.001 *	15.0 (CI95: 10.3–19.7)	0.446
	Unmethylated	13.9 (CI95: 10.7–17.0)		11.2 (CI95: 8.0–14.4)	
TMZ + RTX	MGMT methylated	28.5 (CI95: 20.5–36.4)	<0.001 *	18.6 (CI95: 12.8–24.4)	0.088
	MGMT unmethylated	14.1 (CI95: 10.9–17.3)		11.5 (CI95: 8.1–14.9)	
TMZ + RTX or TMZ alone	MGMT methylated	28.5 (CI95: 20.5–36.4)	<0.001 *	16.3 (CI95: 11.3–21.4)	0.261
	MGMT unmethylated	14.1 (CI95: 10.9–17.3)		11.5 (CI95: 8.1–14.9)	
Surgical Modality	Resection	21.7 (CI95: 17.6–25.8)	0.056	13.9 (CI95: 10.6–17.1)	0.070
	Biopsy	9.5 (CI95: 4.0–15.1)		7.6 (CI95: 3.1–12.0)	
Preoperative Performance	ECOG = 0	22.4 (CI95: 17.7–27.2)	0.146	10.9 (CI95: 8.0–13.9)	0.110
	ECOG ≥ 1	15.7 (CI95: 10.3–21.0)		16.0 (CI95: 10.5–21.5)	

PFS—progression-free survival, MGMT—O6-Methylguanine-DNA-Methyltransferase, TMZ—temozolomide, RTX—radiotherapy, ECOG—Eastern Co-operative of Oncology Group. * Differences are considered as statistically significant if *p* < 0.05.

**Table 3 cancers-14-06180-t003:** Mean estimated OS in comparison between older (more than 65 years) and younger patients (18 to 65 years) in relation to MGMT promotor methylation status, adjuvant therapy, surgical modality and preoperative ECOG.

Variable		18–65 Years, OS in Months	*p*	>65 Years, OS in Months	*p*
MGMT promotor	Methylated	39.0 (CI95: 28.6–49.4)	<0.001 *	16.5 (CI95: 10.6–22.4)	0.373
	Unmethylated	17.0 (CI95: 13.4–20.4)		10.0 (CI95: 7.5–12.4)	
TMZ + RTX	MGMT methylated	45.7 (CI95: 34.3–57.0)	<0.001 *	27.2 (CI95: 17.5–36.9)	0.015 *
	MGMT unmethylated	17.8 (CI95: 14.1–21.4)		12.3 (CI95: 9.2–15.3)	
TMZ + RTX or TMZ alone	MGMT methylated	43.6 (CI95: 32.5–54.7)	<0.001 *	21.3 (CI95: 11.4–23.5)	0.190
	MGMT unmethylated	17.8 (CI95: 14.1–21.4)		12.3 (CI95: 9.2–15.3)	
Surgical modality	Resection	30.0 (CI95: 24.0–36.0)	<0.001 *	17.0 (CI95: 12.3–21.7)	<0.001 *
	Biopsy	7.4 (CI95: 4.2–10.5)		4.6 (CI95: 3.0–6.2)	
Preoperative Performance	ECOG = 0	28.7 (CI95: 22.4–35.1)	0.471	11.3 (CI95: 8.7–13.9)	0.553
	ECOG ≥ 1	21.8 (CI95: 14.1–29.5)		13.9 (CI95: 8.6–19.3)	

PFS—progression-free survival, MGMT—O6-Methylguanine-DNA-Methyltransferase, TMZ—temozolomide, RTX—radiotherapy, ECOG—Eastern Co-operative of Oncology Group. * Differences are considered as statistically significant if *p* < 0.05.

## Data Availability

The raw data were generated in the authors’ institution. The data that support the findings of this study are available on reasonable request from the corresponding author. The data are not publicly available, as they contain information that could compromise the privacy of research participants.

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
