# Peer review of "Age-Dependent Impact of Concomitant Radio-Chemotherapy and MGMT Promotor Methylation on PFS and OS in Patients with IDH Wild-Type Glioblastoma: The Real-Life Data"

_cancers, 2022, doi:10.3390/cancers14246180_

Round 1

Reviewer 1 Report

Initially, the authors stated that biological and not chronological age and performance have more influence on decision-making by patients with malignant brain tissue tumors. They performed a well-designed study aiming to explore how more aggressive therapy procedures result in the life expectancy (progression-free survival (PFS); overall survival (OS)) of primarily older IDH wildtype GBM patients, thus questioning the legitimacy of the generally accepted recommendations for the reduced treatment protocols within such a cohort of patients.

The study is generally well-planned and performed, and the results support conclusions. This study presented sound results that suggest that, in real life, genetically favorable features of the tumor (such as MGMT promoter methylationpositively influence a patient’s survival only when more aggressive therapy is applied. Thus, the authors found that the MGMT promoter methylation manifested a prognostic impact on OS exclusively when associated with adjuvant treatment (concomitant radio-chemotherapy). The main weakness of this study is its retrospective design, which could result in incomplete data. Nevertheless, the authors took steps to avoid possible limitations. They did not evaluate pre-existing specific co-morbidities separately and selected only patients suitable for intracranial surgery under general anesthesia. However, they also emphasized the need for further prospective validation of their observations.  

Minor typographical corrections should be performed:

Line 68;  insert the hyphen between „progression“ and „free“

Line 105; change „was“ to „were“

Line 115; Insert article „The“ before the word „Declaration“

Line 132; Insert the hyphen between „Board“ and „selected“

Line 218; Remove the hyphen between „Age“ and „groups“

Line 218: Change „in concordance to“ to „in concordance with“

Line 233; Insert the hyphen between „Progression“ and „free“

Line 255;  Change „was“ to „were“

Line 287; Correct „O-6-methylguanin-DNA-methyltransferase” to O-6-methylguanine-DNA-methyltransferase”

Line 289;  Change „methylguanin” to methylguanine”

Line 289; Change „guanin” to guanine”

Line 307; Insert the hyphen between „event“and „free“

Author Response

Dear Reviewer #1,

we are very grateful for your reasonable comments considering our manuscript. We were glad to carefully modify all linguistical and typographical corrections exactly as it was recommended.

We thank you one more time for your remarks and are ready to provide additional information.

With regards,

A. Krigers

Reviewer 2 Report

Review Report

Title: Age-dependent impact of concomitant radio-chemotherapy and MGMT promotor methylation on PFS and OS in patients with IDH wild-type glioblastoma: the real-life data.

Having in mind that population worldwide is aging, this study highlights the existing gaps in understanding the difference between chronological and biological age and potential age-biases in the choice of treatment for elderly glioblastoma (GBM) patients. It is estimated that by the year of 2030, 1 in 6 people will be aged (60 years old or even older) and, as the authors of this study indicate, 70 % of all malignancies will be existed in patients older than 65 years. This study emphasizes that the peak incidence for malignant glioblastoma has already reached 75 to 84 years. On the other side, improvements in health status, cognition, and functional independence in older persons or as the authors referred as increased “disability-free life span”, has led some scientists to take into consideration changing cut-off age from 65 years to 75 years to define ‘elderly’, while others recommended use of biological age for defining being ‘old’. In the context of cancer (especially, malignant such as glioblastoma in this study), there is obvious lack of universal guidelines to determine the biological age of the patients and other factors need to be considered to define the most appropriate treatment. For example, age-related changes in DNA methylation and telomerase may be promising molecular markers not only for aging monitoring and predicting life expectancy, but also for the choice of treatment for cancer patients and prediction of survival. Unfortunately, most clinical trials for novel therapeutic agents against cancer exclude elderly patients based on their chronological age and frailty. Besides, Stupp’s regimen, the most successful clinical protocol for glioma, combines radiotherapy with concomitant temozolomide (TMZ) followed by up to six cycles of adjuvant TMZ, was not tested on patients older than 65 years due to conservative age cut-offs in previous clinical trials, as the authors underlines.  Therefore, elderly GBM patients with MGMT  methylated promoter were allowed to receive adjuvant TMZ monotherapy MGMT  methylation while other older patients were treated with hypo-fractioned radiotherapy. Regardless of the choice of therapy both groups of older GBM patients had the worse outcomes compared with younger patients treated by Stupp’s protocol. The study in front of us valued biological, not chronological, age, including frailty, in decision-making in older GBM patients and allowed neurologically safe maximum tumor resection followed by concomitant radio-chemotherapy (Stupp’s regimen). Study was conducted in a large cohort of 273 firstly diagnosed IDH wild-type glioblastomas in which 142 (52%) patients were older than 65 years. Notably, concomitant radio-chemotherapy provided survival advantages in both younger and older group. Also, positive impact of the methylated MGMT promoter to overall survival (OS) in GBM patients receiving  concomitant radio-chemotherapy is observed. Results are well presented and disscussed and conclusions are clear and concise.

The significance of this study is that for the first time aggressive treatment for malignant GBM in elderly is applied and investigated and significant progress in OS is determined (20.0 months). Therefore, I warmly recommend this manuscript for publishing in your journal after minor correction of following errors:

Line 76: “Hence, this provided a representable taxonomic cohort in accordance to the current 5th WHO classification of tumors of the central nervous system from 2021[10].”

should be corrected as:

 “Hence, this provided a representable taxonomic cohort in accordance with the current 5th WHO classification of tumors of the central nervous system from 2021 [10].”

***

Line 115: “This study was conducted in accordance to Declaration of Helsinki and its future amendments.”

should be corrected as:

 “This study was conducted in accordance with Declaration of Helsinki and its future amendments.”

***

Line 150: “PFS in accordance to MGMT promotor methylation status,…”

should be corrected as:

“PFS in accordance with MGMT promotor methylation status,…”

***

Line 218: “Preoperative performance score or surgical modality did not provide additional PFS hazards in both age-groups in accordance to Cox regression.”

should be corrected as:

“Preoperative performance score or surgical modality did not provide additional PFS hazards in both age-groups in accordance with Cox regression.”

***

Line 261: “Generally, the population is getting older and the disability-free life span is increased [4]. ”

 should be corrected as:

“Generally, the population is getting older, and the disability-free life span is increased [4]. ”

Author Response

Dear Reviewer #2,

We are delighted that you found an opportunity and time to read and review our manuscript. We were glad to carefully modify all linguistical and typographical corrections exactly as it was recommended.

We thank you one more time for your remarks and are ready to provide additional information.

With regards,

A. Krigers

Reviewer 3 Report

I think this is an important study to determine future treatment strategies for glioblastoma.

I had difficulty understanding the first two sentences of the results. Regarding the number of patients, it was difficult to understand the explanation why 163 + 129 = 292 and not 273.

The NCCN guidelines divide the treatment plan into two categories: under 70 years of age and the rest of the population, but why does this study divide the patients into two categories: under 65 years of age or the rest of the population?

After dividing the cases by age, I believe the cases are divided by ECOG and KPI, and then the treatment plan is decided. In this study, it was difficult to understand whether the cases were classified by ECOG or KPI. I would like to see a comparison of treatment effects by age and ECOG or KPI scores, but it was difficult to understand.

Author Response

Dear Reviewer #3,

we are very grateful for your support and time in improving our manuscript. Without any doubts, your comments made the manuscript better. We were glad to carefully modify our text according or provide additional explanation to all your concerns. Further, we provide our clarification and applied changes more detailed.

1. I had difficulty understanding the first two sentences of the results. Regarding the number of patients, it was difficult to understand the explanation why 163 + 129 = 292 and not 273.

Thank you for this remark. There was an accidental data transfer failure considering gender stratification. We corrected this aspect. Moreover, we re-calculated other descriptive data to be sure that it was the only point.

  • We modified the second sentence of the results:“Mean age of the 155 (57 %) males and 118 (43 %) females was 64 years (CI95: 62 – 66).”

2. The NCCN guidelines divide the treatment plan into two categories: under 70 years of age and the rest of the population, but why does this study divide the patients into two categories: under 65 years of age or the rest of the population?

The definition of ‘elderly’ is not strictly defined and various cut-offs are internationally used. In SNO and EANO guidelines, both 65 and 70 years as cut-off are provided and even sometimes showed as interval. We used 65-years of age as cut-off like numerous previous neuro-oncological trials (Perry at all 2017, EF-14, NOA-08, PARADIGM). WHO defines ‘elderly’ as older than 65 years as well. The CBTRUS database specifies age-interval as 65-75 years. Moreover, the mean age of our cohort was about 65 years, and this cut-off is set as our internal clinical standard.

3. After dividing the cases by age, I believe the cases are divided by ECOG and KPI, and then the treatment plan is decided. In this study, it was difficult to understand whether the cases were classified by ECOG or KPI. I would like to see a comparison of treatment effects by age and ECOG or KPI scores, but it was difficult to understand.

In our center, KPI as performance scale is primary used for the treatment strategy decision (as provided in methods). For retrospective binominal performance comparison, we calculated ECOG for all patients according to internationally used criteria (Oken at all), e.g., ECOG 0 in concordance with KPI 90-100 vs. ECOG ³1 in concordance with KPI ≤80.

The PFS and OS stratification is correspondingly provided in table 2 and table 3. As the last point of each table, the descriptive and analytical evaluation of PFS and OS considering performance (ECOG 0 vs. ECOG ³1) and simultaneously age (18-65 vs. ³65 years) is provided.

4. Extensive editing of English language and style required 

Extensive linguistic and typographical review was performed by a native speaker one more time, and corresponding corrections were applied.

We thank you one more time for your remarks and are ready to provide additional information.

With regards,

A. Krigers